# Influence of Foot Morphology on the Center of Pressure Pattern in Patients with Down Syndrome

**DOI:** 10.3390/ijerph20042769

**Published:** 2023-02-04

**Authors:** Cristina Ferrario, Claudia Condoluci, Marco Tarabini, Carlotta Maria Manzia, Gabriella Di Girolamo, Massimiliano Pau, Manuela Galli

**Affiliations:** 1Department of Mechanical Engineering, Politecnico di Milano, 20133 Milano, Italy; 2Department of Electronics, Information of Bioengineering, Politecnico di Milano, 20133 Milano, Italy; 3IRCCS San Raffaele Pisana, Via della Pisana 235, 00163 Rome, Italy; 4Department of Mechanical, Chemical and Materials Engineering, University of Cagliari, 09123 Cagliari, Italy

**Keywords:** down syndrome, flat foot, postural balance, gait disorders

## Abstract

Background: The primary aim of this study was to assess how different conformations of the foot in individuals with Down syndrome affected the CoP during walking, and the secondary aim was to evaluate the effect of an excess of mass in young adults and children with Down syndrome and flat foot. The greater investigation of these aspects will allow for more targeted rehabilitation treatments to improve a patient’s quality of life. Methods: The tests were carried out on 217 subjects with Down syndrome, 65 children and 152 young adults, and on 30 healthy individuals, 19 children and 11 young adults. All subjects underwent gait analysis, and the group with Down syndrome was also assessed with baropodometric tests to evaluate foot morphology. Results: The statistical analysis showed that within both the young adult and child groups, the CoP pattern in the anterior–posterior direction reflected a difficulty in proceeding in the walking direction compensated by a medio–lateral swing. The gait of children with Down syndrome was more impaired than that of young adults. In both young adults and children, a higher severity of impairment was found in overweight and obese female individuals. Conclusions: These results suggest that the sensory deficits and the development of hypotonic muscles and lax ligaments of the syndrome lead to morphological alterations of the foot that, combined with the physical characteristics of short stature and obesity, negatively impact the CoP pattern of people with Down syndrome during walking.

## 1. Introduction

Down syndrome (DS), also referred to as “Trisomy 21”, is the most common chromosomal abnormality that causes the presence of a third copy (all or part of it) of chromosome 21 [1,2]. DS is generally associated with impaired muscle function [3,4] and joint laxity [5], leading to debilitating gait disturbances that vary from patient to patient [6]. In particular, these individuals are often characterized by musculoskeletal disorders of the foot such as hallux valgus, syndactyly, clinodactyly and flat foot. The last issue, which has been estimated to affect up to 60–80% of individuals in childhood and adolescence [7,8], is mainly caused by muscular hypotonia and ligamentous laxity, two key factors that can favor the collapse of the longitudinal arch of the foot [1,5,8,9].

Since flat feet can significantly alter several aspects of the gait pattern of individuals with DS [9], the use of quantitative, computerized gait analysis represents a very useful tool to elucidate the mechanisms with which features related to this genetic condition differentiate the gait of people with DS from unaffected individuals. For example, it has been observed that the flat-footed condition affects the generated power of the ankle, leading to muscle fatigue and an energetically more expensive gait compared with those of individual with DS with a normal arch or healthy subjects [9,10].

However, it is noteworthy that little information is available regarding the trajectory of the Center of Pressure (COP, the point of application of the ground reaction force (GRF)) during the gait of individuals with DS. The aim of this study was therefore to bridge this gap in order to assess the effects of a morphological change in the foot on gait. In particular, to our knowledge, no studies have investigated the existence of a possible link between a flat foot and the alteration of the trajectory of the CoP during walking. On the basis of these considerations, this study was aimed to assess how different conformations of the feet in individuals with DS affect the displacement and velocity of the CoP during walking. As secondary goal, we intended to verify if, and to what extent, a mass excess (which is typical of individuals with DS) can further influence the COP pattern in young adults and children with DS and flat feet. We expected to observe a greater impairment in children than in young adults because of the compensatory strategies that patients acquire with age that children do not yet possess [11]. In addition, we expected that there would be variation related to the sex of the participants, as different body structures alter the biomechanics of movement and females tend to have a lower CoM than males. Another parameter that we expected to influence the results was the weight of the participants, as overweight and obesity have been found to worsen people’s biomechanics.

By assessing the biomechanics of movement by means of spatiotemporal, kinetic, and dynamic parameters concerning the gait of people with morphological alterations of the feet, it is possible to direct a physician, or rehabilitation therapist, toward the choice of the best aid or, in general, the best rehabilitation treatment for their patient. Flat feet represents a particular foot conformation that impacts the distribution of GRF during walking, and it has been demonstrated to cause muscle fatigue and pain. DS individuals with flat feet have a tendency to excessively rotate the foot, and this behavior makes it difficult to generate energy during the pushing phase. For this reason, rehabilitative treatments are considered advisable in order to reduce foot rotation and stabilize gait; orthotic insoles designed for the purpose of supporting the medial longitudinal arch and reducing foot pronation are often prescribed [12]. Thus, the results of this study could help provide more targeted treatments to the specific needs of patients, and this could mean a significant improvement in the quality of life.

Morphological alternations of the foot are classified according to the geometric and functional characteristics of the plantar arch, and three different foot conformations can be distinguished: the normal foot, the cavus foot, and the flat foot. A quick and useful measure to characterize plantar morphology is the calculation of the arch index of the arch (“Arch Index”). Proposed by Cavanagh and Rodgers in 1987, the arch index is a numerical value that can be obtained from the following procedure (Figure 1) [13]:A line is considered that connects the center of the ilex toe to the center of the heel of the foot.One divides the axis obtained by identifying the three regions of the foot into three identical parts: hindfoot (A), midfoot (B) and forefoot (C).The Arch Index is the ratio between the surface area of the midfoot and the total surface area of the foot (excluding toes).

Based on the obtained Arch Index value, a foot can be defined as cavus if the index is less than or equal to 0.21, normal if it is between 0.21 and 0.26, or flat if it is greater than or equal to 0.26 [13].

## 2. Materials and Methods

### 2.1. Participants

Individuals with DS were examined at the Movement Analysis Lab of the IRCCS “San Raffaele Pisana” (Rome, Italy). From this population, a sample of 65 children (DS_c: aged 12 and less) and 152 adolescents and young adults (DS_a: over 12 years) with DS (pure trisomy 21 chromosomal anomaly) was selected for the present study. Subjects with DS were divided into young adults and children in order to assess whether age was a factor influencing the performed analyses. Thirty unaffected individuals were recruited from the Movement Analysis Lab Luigi Divieti (Milan, Italy) and served as a control group (CG). The CG consisted of 19 children (CG_c) and 11 young adults (CG_a). Inclusion criteria for DS were the absence of previous major surgery or orthopedic treatment. In addition, patients with clinical signs of dementia were excluded, and, to be eligible for testing, patients had to be able to walk independently without the assistance of a physiotherapist or the use of supportive devices. For the CG, cardiovascular, musculoskeletal or neurological disorders were used as exclusion criteria. A written informed consent form containing all details of the study was signed by all involved participants.

The anthropometric characteristics of the groups are detailed in Table 1.

### 2.2. Experimental Protocol

All participants underwent a 3D computerized gait analysis, which was carried out using an optical motion capture system (ELITE2002, BTS, Milan, Italy) composed of 12 cameras set at a frequency of 100 Hz. Prior to the tests, a number of anthropometric measures (height, weight, anterior superior iliac spine distance, pelvis thickness, knee and ankle width, and leg length) for each participant were acquired. Then, 22 spherical retro-reflective passive markers (14 mm diameter) were placed on the skin of lower limbs and trunk at particular landmarks following the protocol described by Davis et al. [14]. Participants were required to walk at a comfortable self-selected speed along a 10 m walkway six times.

By using two force platforms (Kistler, CH), the displacement of the CoP and the GRF components in antero–posterior (AP), vertical (V) and medio–lateral (ML) directions were collected for each trial. The stance phase was identified by considering the GRF-V pattern and its two peaks found at 11% and 49% of the stride cycle [15]. The parameters that were extracted in this phase were: (i) AP, the ML excursion of the CoP; (ii) the coefficient of determination, r^2^, in the AP and ML directions; (iii) the slope of the line interpolating the CoP’s displacement section in the AP and ML directions; and (iv) the average AP and ML velocity of the CoP.

The calculation of the excursion along the AP and ML directions was performed by subtracting the absolute values of the maximum and minimum CoP coordinates. The values of the average CoP velocities in the AP and ML directions were obtained from the calculation of the slopes of the CoP displacement curves.

For the analysis, only velocity, and thus slope, values obtained from graphs that could be interpolated as straight lines, i.e., with a coefficient of determination (r^2^) greater than 0.7, were considered valid. However, the r^2^ parameter was also included in the statistical analysis in order to observe whether there were any differences in the linear trend of the CoP between the considered groups. All data were analyzed with the BTS SMART Analyzer.

Baropodometric tests were carried out for the group of DS subjects only to identify their foot morphology (i.e., flat, cavus or normal) by calculating the Arch Index (AI) as described before [13]; this was done because the CG did not present gait alterations regardless of the morphology of the foot, so no differentiation was made on the basis of the foot in the CG. Plantar pressure measurements were performed using a pressure-sensitive mat (Tekscan Inc., South Boston, MA, USA) consisting of sensors arranged in a 42 × 48 matrix [8].

### 2.3. Statistical Analysis

All statistical analyses were performed using Minitab software (Statistical Software, GMSL, Nerviano, Italy).

As a first step, a statistical analysis was performed to compare the anthropometric data. First, the normality of each sample was checked by means of the Kolmogorov–Smirnov test with a significance level of 0.05. In the case of a normal distribution, the comparison was conducted by using the two-sample *t*-test; otherwise, the Mann–Whitney U Test was applied.

Secondly, the same statistical analysis was used to compare the parameters of interest extracted from the CoP pattern graphs in the AP and ML directions. In particular, Bonferroni’s method was used to conduct multiple comparisons in each of the two groups, considering the division of the CG according to sex and that of the DS group according to both sex and different foot conformations.

Finally, in order to observe whether weight could have an influence, a further statistical analysis was performed to compare the various groups divided into obese (BMI adult ≥ 30; BMI children > 97th percentile), overweight (25 ≤ BMI adult < 30; 75th ≤ BMI children < 97th percentile), and normal weight (BMI adult < 25; BMI children < 75th percentile) categories. In order to have a consistent number of subjects, this analysis was only carried out in two groups: DS_c and DS_a, both with flat feet. The Bonferroni method was used to carry out multiple comparisons in each of the two groups, considering the division of the subjects according to both sex and weight class (normal weight, overweight, and obese).

## 3. Results

### 3.1. Participants

As found in the literature [7,15,16,17], a considerable percentage of individuals with DS are affected by generally severe flat feet. In the DS_c group, a flat foot incidence of 82.4%, a normal foot incidence of 7.6%, and a cavus foot incidence of 10% were observed. In the DS_a group, a flat foot incidence of 80.4%, a normal foot incidence of 13.2%, and a cavus foot incidence of 6.4% were calculated. The anthropometric characteristics of the groups are detailed in Table 1, where normality was checked using the Kolmogorov–Smirnov test and then Student’s *t*-test or nonparametric Mann–Whitney test were used to compare means.

The initial statistical analysis of the anthropometric data revealed some statistically significant differences between the DS_c and CG_c groups and between the DS_a and CG_a groups. Considering the comparison between the children, the height of the DS_c group was significantly lower than that of the CG_c group (*p*-value = 0.001). The same result was obtained when comparing the height of DS_a with that of CG_a (*p*-value = 0.003). The DS_a group also showed a significantly higher BMI than the CG_a group (*p*-value = 0.02 10^−2^).

As is already known, individuals with DS are normally shorter and have a higher body weight than healthy individuals due to the disease from which they suffer [18,19]. Therefore, despite the differences in our anthropometric measures of height and BMI, it was considered possible to compare the groups because these differences are characteristic of DS [20,21].

Specifically, having found a wide variability in height between subjects, the values of the mean CoP velocity were normalized with respect to height, as already performed in the literature [22,23].

### 3.2. Children’s Group

The mean and standard deviation values of the parameters collected from the gait analysis of the children’s group are shown in Table 2.

Statistical analysis showed no statistically significant differences from the sex division, so the data were considered to belong to the same group.

Statistically significant differences were found between the CG_c and DS_c groups. Higher Ang AP, v/h AP, and AP range values were observed in the CG_c group than in the DS_c Normal (*p*-value < 0.001; *p*-value = 0.02 × 10^−2^; *p*-value < 0.001), DS_c Flat (*p*-value < 0.001; *p*-value < 0.001; *p*-value < 0.001), and DS_c Cavus (*p*-value < 0.001; *p*-value < 0.001; *p*-value < 0.001) groups. The CG_c group also showed a more linear CoP trend in the AP direction, and thus a value of r^2^ AP closer to 1, than that observed in the DS_c Normal (*p*-value = 0.02) and DS_c Flat (*p*-value = 0.02) groups.

Regarding the parameters in the ML direction, the Ang ML was greater in the CG_c group than that observed in the DS_c Flat (*p*-value < 0.001), DS_c Normal (*p*-value = 0.03), and DS_c Cavus (*p*-value = 0.01) groups. In addition, significantly higher mean values of ML range and v/h ML were observed in CG_c than in DS_c Flat (*p*-value = 0.03 × 10^−2^; *p*-value < 0.001).

### 3.3. Young Adult Group

The mean and standard deviation values obtained for the young adult group are shown in Table 3.

Statistically significant differences were found both between the healthy group and subjects with DS and within the group of DS subjects with different foot morphologies. The DS_a Normal group showed a less linear trend in the AP direction of the CoP, and thus a value of r^2^ AP closer to 0, than both the DS_a Flat group (*p*-value = 0.001) and the CG_a group (*p*-value = 0.01); in particular, r^2^ AP values were lower in the DS_a Normal group. Considering the sex division, lower r^2^ AP values were recorded in women in the DS_a Normal group than in women in the DS_a Flat group (*p*-value = 0.001). In addition, the Ang AP, AP range and v/h AP values were higher in the CG_a group than in the DS_a Normal (*p*-value = 0.001; *p*-value < 0.001; *p*-value = 0.03) and DS_a Flat (*p*-value = 0.001; *p*-value < 0.001; *p*-value = 0.04) groups. The AP range was also significantly greater in the CG_a group than in the DS_a Cavus group (*p*-value = 0.004). Considering the ML direction, statistically significant differences emerged within the DS_a group: in the DS_a Flat group, women showed lower ML range values than men (*p*-value = 0.002).

### 3.4. Children and Young Adults with Down Syndrome and Flat Feet

The DS Flat group was considered for further analysis by adopting the subdivision of subjects according to the body mass index. Initially, a descriptive analysis was performed to calculate the mean and standard deviation values of each parameter for the DS_c Flat group and the DS_a Flat group, considering the division of the subjects according to sex and weight class (normal weight, overweight, and obese). By means of Bonferroni post-hoc analysis, multiple comparisons were carried out in order to search for statistically significant differences within the young adult and child groups.

The mean and standard deviation data obtained for the DS_c Flat group are shown in Table 4.

Statistically significant differences emerged regarding the Ang AP, AP range and v/h AP parameters. Obese females showed significantly lower values than overweight females (*p*-value = 0.001; *p*-value < 0.001; *p*-value = 0.01), normal weight males (*p*-value = 0.001; *p*-value = 0.001; *p*-value = 0.001), overweight males (*p*-value < 0.001; *p*-value < 0.001; *p*-value = 0.01), and obese males (*p*-value = 0.01; *p*-value = 0.01).

The mean and standard deviation values obtained for the DS_a Flat group are shown in Table 5.

Statistically significant differences were found between men and women with regard to the parameters of v/h AP and, as already observed from the comparison within the young adult group, ML range; in women, higher mean v/h AP values (*p*-value = 0.002) and lower ML range values (*p*-value < 0.001) were recorded than in men. In particular, the ML range was greater in obese men than in obese (*p*-value = 0.03) and overweight women (*p*-value = 0.002), and it was greater in normal weight men than in overweight women (*p*-value = 0.04). Lastly, obese young adults generally showed a less linear CoP trend in the AP direction than overweight young adults: r^2^ AP values were statistically lower in the first group (*p*-value = 0.03).

## 4. Discussion

### 4.1. Comparison between Healthy and DS Subjects

Within both the young adult and child groups, statistically significant differences were found between DS and healthy subjects, regardless of foot morphology. In particular, in the young adult group, the r^2^, Ang, range, and v/h parameters in the AP direction were lower in DS subjects than in the healthy group. In the children’s group, statistically significant differences were also observed for Ang, range and v/h parameters in the ML direction; these were lower in DS children than in healthy children.

The obtained results suggest the existence of a peculiar gait pattern in individuals with DS. This pathology involves sensory deficits and the development of hypotonic muscles and lax ligaments that often lead to the development of morphological alterations of the foot, which, combined with the physical characteristics of short stature and obesity, impact the CoP pattern during walking. In particular, the CoP displacement pattern in the AP direction was less linear and had a smaller slope in subjects with DS; this result, in addition to the variations found in the v/h and range parameters in the AP direction, reflects a difficulty in proceeding in the walking direction. It is known that individuals with DS tend to cope with their sensory and physical deficits by adopting compensatory mechanisms that alter gait parameters [24]: in particular, they swing in the ML direction by increasing stride width and decreasing their velocity during the single limb stance phase to improve the stability of movement [2,23,25]. Consequently, there are decreases in the v/h and CoP range in the AP direction.

Observing statistically significant differences in CoP parameters in the ML direction additionally suggested that compared with that of the CG, the gait of children with DS is more impaired than that of young adults with DS. As found in the literature, the movements of DS subjects appear clumsy and uncoordinated, but they often tend to improve over the years, although performance is still lower than that of healthy subjects [26,27,28,29,30]. The improvement that was seen in young adults compared with the children’s group seems to be imputable to the compensatory strategies that subjects develop over the years to reduce energy expenditure during walking.

### 4.2. Comparison within the Young Adult Group

Considering the movement analysis of the young adult group, the parameter identified to characterize the morphology of the foot in DS was the r^2^. In fact, with the same sex, the r^2^ coefficient in the AP direction was greater in women with DS with flat feet than those with normal feet. It seems that this difference was due to the rigidity of the flat feet, which tend to lose their ability to adapt to the ground and to be less flexible during the phase of full plantar support. As a result, the CoP pattern was more linear along the direction of walking, showing a low variability over time. This altered stiffness of the feet also led to a difference in the GRF, thus affecting the subject’s propulsion phase and advancement in the direction of walking.

Considering the ML range, however, sex differences were found within the group of young adults with DS. With the same foot morphology, it was observed that this parameter tended to be lower in DS women with flat feet than in DS men with flat feet. This could have been due to the fact that women tend to stiffen their feet more in the full stance phase and thus apply a greater vertical force, leading to a lower ML range. Women, in particular, tend to have a body conformation that places the CoP in a different position than men, so in attempting to move their CoP forward while walking, they tend to apply a greater vertical force. This behavior may be reflected in a lower ML CoP range.

### 4.3. Comparison within the Group of DS Young Adults with Flat Feet

When analyzing the group of young adults with DS and flat feet in detail, the ML range was lower in obese women than in obese men and in overweight women than in normal weight and obese men. These results confirmed the presence of differences in the ML range due only to sex and not to weight class [31].

Accordingly, it was observed that the CoP’s v/h in the AP direction tended to be greater in women than in men; it appears that applying a greater vertical ground reaction force not only resulted in a lower range ML but also tended to facilitate the CoP’s advancement in the AP direction.

Continuing with the analysis within this group, the r^2^ AP parameter in obese young adults was statistically lower than that in overweight young adults. This result suggested that obesity in DS young adults with and flat feet has an impact on the CoP pattern in the AP direction during the single limb stance phase. Thus, it appears that a high BMI leads to a more pronounced clumsiness that results in a more difficult gait and greater anomalies in the CoP pattern in the AP direction [32].

### 4.4. Comparison within the Group of DS Children and Flat Feet

When analyzing the group of DS children with flat feet in detail, statistically significant differences emerged for the Ang parameter in the AP direction; this was lower in obese girls than in obese boys. The same differences were also found for the range and v/h parameters of the CoP in the AP direction. Even in this case, the differences found between boys and girls could be attributed to the different body morphology of girls compared with that of boys; the downward shift of the CoP seems to result in greater difficulty in the progression of the body in the AP direction, as found in the young adult group. Lower range and Ang values in the AP direction are indicative of less flexibility of movement in this direction. In particular, the Ang AP in obese DS girls with flat feet was the lowest value found among all groups studied in this research, including healthy subjects and those with DS and considering differences in age, sex, BMI and foot morphology. The value of this parameter was 4°, which reflects the difficulty of obese DS girls with flat feet to proceed in the direction of walking.

Though a higher v/h AP value was recorded for DS women with flat feet than for DS men with flat feet, this parameter was lower in DS girls with flat feet than in DS boys with flat feet. Contrary to women, a significantly elevated vertical force component was not observed in DS girls with flat feet, so it appears that girls have more difficulty compensating for the ML force component and, in general, for balance disorders due to their body conformation.

As in DS young adults with flat feet, obesity had an impact on the CoP pattern in the AP direction during the stance phase. Regarding the Ang AP, AP range and v/h AP parameters, lower values were recorded in obese girls than in overweight girls [33].

In accordance with findings in the literature, the anomalies observed in obese DS children may be associated with poor motor skill development due to early obesity [19,20,21]. Obesity in DS children may lead to increased levels of oxidative stress, resulting in a variation in the musculature of the lower limbs and difficulty in advancing in the AP direction. An excess of subcutaneous adipose tissue also indicates reduced joint mobility, and the combination of these factors may result in more clumsy and less coordinated movements, reflecting the variations found in the CoP pattern in the AP direction [16,20,34,35].

## 5. Conclusions

Having observed greater differences between DS and healthy children than between DS and healthy young adults in this study, it appears that the gait of DS children is more impaired than that of young adults. Compared with healthy young adults, DS young adults did not show any noticeable abnormalities in the ML direction, in contrast to DS children compared with the healthy group; these results suggest that some abnormalities in the movements present in children are reduced over time.

The most indicative results were obtained from comparisons made within the young adult DS group with flat feet and within the child DS group with flat feet. Having calculated an incidence of flat feet of around 80% in both the groups of DS children and DS young adults, it appears that this morphological alteration is not age-related but a specificity of DS.

According to the obtained results, the presence of flat feet in the young adults with DS differentiated behavior in women and men in terms of the CoP range in the ML direction and v/h in the AP direction and in obese and overweight subjects in terms of r^2^ AP. In the group of DS children, the presence of flat feet also differentiated movement in females and males and in obese and overweight subjects in terms of Ang, range, and v/h in the AP direction.

These differences within the group of DS subjects with flat feet suggest that sex and BMI are parameters that influence the movement of these subjects. It is therefore considered appropriate that the choice of the footbed, or rehabilitation treatment in general, should consider the differences in the body conformation of males and females and of normal, overweight, and obese subjects.

The only parameter that was able to characterize foot morphology in DS turned out to be r^2^ AP, although the comparison concerned the group of DS women with normal feet, which had a small amount of data. In conclusion, the CoP pattern during walking in DS subjects seems to be influenced by morphological alterations of the feet and by sex and BMI characteristics.

## Figures and Tables

**Figure 1 ijerph-20-02769-f001:**
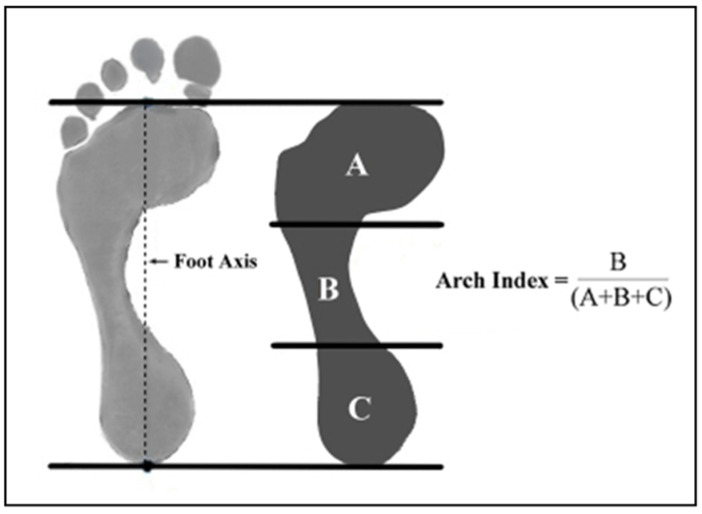
Calculation of the Arch Index.

**Table 1 ijerph-20-02769-t001:** Mean and (standard deviation) values of anthropometric data of the control and DS groups divided into children and young adults. M, males; F, females; BMI, body mass index; AI right, arch index of the right foot; AI left, arch index of the left foot; * *p*-value < 0.05 when comparing the control and DS groups.

	Sex	Age	Height (cm)	Weight (kg)	BMI (kg/m^2^)	AI Right	AI Left
DS_c	38 M 27 F	9.63 (1.70)	127.5 (12.1) *	34.8 (11.9)	20.8 (4.03) *	0.31 (0.08)	0.32 (0.07)
DS_a	82 M 70 F	21.86 (8.95)	148.0 (9.87) *	56.5 (13.5)	25.6 (5.17) *	0.31 (0.06)	0.30 (0.06)
CG_c	14 M 5 F	9.00 (2.03)	139.8 (13.4) *	39.0 (11.9)	19.4 (3.24) *	-	-
CG_a	4 M 7 F	15.36 (3.08)	161.4 (11.5) *	54.3 (14.7)	20.5 (3.18) *	-	-

**Table 2 ijerph-20-02769-t002:** Mean and standard deviation values of parameters extracted from the CoP in the antero–posterior (AP) and medio–lateral (ML) directions for healthy children and children with Down syndrome (DS). M, males; F, females; r^2^, coefficient of determination; v/h, mean velocity of the CoP normalized with respect to height; Ang, angle of inclination of the line interpolating the displacement of the analyzed CoP; range, range of the CoP (max coordinate–min coordinate).

	DS_c Flat		DS_c Normal		DS_c Cavus		CG_c	
	M	F	M	F	M	F	M	F
r^2^ AP	0.88 (0.08)	0.91 (0.07)	0.85 (0.12)	0.86 (0.08)	0.92 (0.06)	0.88 (0.12)	0.94 (0.05)	0.93 (0.04)
v/h AP (s^−1^)	0.14 (0.06)	0.12 (0.05)	0.15 (0.06)	0.13 (0.05)	0.17 (0.07)	0.10 (0.05)	0.22 (0.05)	0.25 (0.07)
Ang AP (°)	10.3 (4.07)	8.59 (3.65)	9.87 (4.53)	8.68 (3.39)	12.7 (4.40)	6.77 (3.51)	16.8 (3.50)	18.6 (3.69)
AP range (m)	0.07 (0.03)	0.07 (0.03)	0.08 (0.03)	0.06 (0.02)	0.08 (0.02)	0.05 (0.02)	0.11 (0.02)	0.11 (0.02)
r^2^ ML	0.85 (0.08)	0.89 (0.08)	0.84 (0.05)	0.88 (0.07)	0.86 (0.08)	0.87 (0.10)	0.86 (0.09)	0.84 (0.06)
v/h ML (s^−1^)	0.03 (0.02)	0.02 (0.01)	0.04 (0.01)	0.03 (0.03)	0.02 (0.02)	0.05 (0.03)	0.04 (0.02)	0.07 (0.05)
Ang ML (°)	1.99 (1.35)	1.62 (0.69)	3.07 (0.91)	2.10 (1.69)	1.74 (1.39)	3.28 (1.82)	3.13 (1.39)	5.41 (4.01)
ML range (m)	0.01 (0.01)	0.01 (0.01)	0.02 (0.01)	0.01 (0.01)	0.01 (0.01)	0.02 (0.01)	0.02 (0.01)	0.02 (0.01)

**Table 3 ijerph-20-02769-t003:** Mean and standard deviation values of parameters extracted from the CoP in the antero–posterior (AP) and medio–lateral (ML) directions for healthy young adults and young adults with Down syndrome (DS). M, males; F, females; r^2^, coefficient of determination; v/h, mean velocity of the CoP normalized with respect to height; Ang, angle of inclination of the line interpolating the displacement of the analyzed CoP; range, range of the CoP (max coordinate–min coordinate).

	DS_a Flat		DS_a Normal		DS_a Cavus		CG_a	
	M	F	M	F	M	F	M	F
r^2^ AP	0.93 (0.07)	0.92 (0.07)	0.92 (0.07)	0.86 (0.09)	0.94 (0.04)	0.92 (0.06)	0.94 (0.05)	0.96 (0.03)
v/h AP (s^−1^)	0.13 (0.05)	0.14 (0.05)	0.14 (0.05)	0.12 (0.05)	0.15 (0.05)	0.15 (0.07)	0.15 (0.04)	0.19 (0.05)
Ang AP (°)	11.2 (4.27)	11.6 (3.98)	11.7 (4.05)	10.1 (4.03)	12.7 (3.93)	11.9 (5.57)	13.9 (3.26)	16.8 (3.99)
AP range (m)	0.08 (0.03)	0.08 (0.02)	0.09 (0.02)	0.07 (0.02)	0.08 (0.03)	0.08 (0.03)	0.10 (0.01)	0.12 (0.03)
r^2^ ML	0.88 (0.09)	0.88 (0.08)	0.87 (0.09)	0.87 (0.09)	0.88 (0.09)	0.88 (0.08)	0.93 (0.03)	0.84 (0.09)
v/h ML (s^−1^)	0.03 (0.02)	0.02 (0.02)	0.03 (0.02)	0.03 (0.02)	0.03 (0.02)	0.01 (0.01)	0.03 (0.01)	0.02 (0.01)
Ang ML (°)	2.51 (1.79)	1.96 (1.32)	2.88 (1.71)	2.74 (1.39)	2.81 (1.73)	1.09 (0.98)	2.44 (0.77)	2.13 (1.24)
ML range (m)	0.02 (0.01)	0.01 (0.01)	0.02 (0.01)	0.01 (0.01)	0.02 (0.01)	0.01 (0.004)	0.02 (0.01)	0.02 (0.01)

**Table 4 ijerph-20-02769-t004:** Mean and standard deviation values of parameters extracted from the CoP in the antero–posterior (AP) and medio–lateral (ML) directions for children with Down syndrome and flat feet divided into males and females and according to weight class (normal weight, overweight, and obese). r^2^, coefficient of determination; v/h, mean CoP velocity normalized with respect to height; Ang, angle of inclination of the line interpolating the analyzed CoP displacement trait; range, range of the CoP (max coordinate–min coordinate).

	Male			Female		
	Normal Weight	Overweight	Obese	Normal Weight	Overweight	Obese
r^2^ AP	0.86 (0.09)	0.89 (0.06)	0.90 (0.09)	0.90 (0.08)	0.92 (0.07)	0.87 (0.08)
v/h AP (s^−1^)	0.15 (0.07)	0.13 (0.05)	0.15 (0.08)	0.12 (0.04)	0.13 (0.05)	0.06 (0.02)
Ang AP (°)	10.3 (4.66)	10.3 (3.65)	10.5 (4.35)	8.46 (3.10)	9.73 (3.50)	4.19 (1.46)
AP range (m)	0.07 (0.03)	0.07 (0.03)	0.07 (0.02)	0.06 (0.02)	0.08 (0.03)	0.03 (0.01)
r^2^ ML	0.86 (0.06)	0.85 (0.09)	0.86 (0.10)	0.90 (0.07)	0.88 (0.09)	0.90 (0.04)
v/h ML (s^−1^)	0.03 (0.03)	0.02 (0.01)	0.03 (0.02)	0.03 (0.01)	0.02 (0.01)	0.02 (0.002)
Ang ML (°)	2.05 (1.67)	1.88 (1.16)	2.32 (1.58)	1.74 (0.83)	1.57 (0.67)	1.66 (0.16)
ML range (m)	0.01 (0.01)	0.01 (0.01)	0.01 (0.01)	0.01 (0.01)	0.01 (0.01)	0.01 (0.01)

**Table 5 ijerph-20-02769-t005:** Mean and standard deviation values of parameters extracted from the CoP in the antero–posterior (AP) and medio–lateral (ML) directions for young adults with Down syndrome and flat feet divided into males and females and according to weight class (normal weight, overweight, and obese). r^2^, coefficient of determination; v/h, mean CoP velocity normalized with respect to height; Ang, angle of inclination of the line interpolating the analyzed CoP displacement trait; range, range of the CoP (max coordinate–min coordinate).

	Male			Female		
	Normal Weight	Overweight	Obese	Normal Weight	Overweight	Obese
r^2^ AP	0.93 (0.06)	0.95 (0.07)	0.91 (0.07)	0.91 (0.07)	0.93 (0.06)	0.92 (0.07)
v/h AP (s^−1^)	0.14 (0.06)	0.12 (0.05)	0.12 (0.04)	0.14 (0.06)	0.15 (0.05)	0.14 (0.05)
Ang AP (°)	11.6 (4.73)	10.8 (3.87)	10.6 (3.46)	11.4 (4.39)	12.0 (3.77)	11.4 (3.84)
AP range (m)	0.08 (0.03)	0.07 (0.03)	0.09 (0.02)	0.08 (0.02)	0.09 (0.02)	0.08 (0.02)
r^2^ ML	0.87 (0.09)	0.88 (0.09)	0.89 (0.07)	0.87 (0.08)	0.88 (0.08)	0.88 (0.09)
v/h ML (s^−1^)	0.03 (0.02)	0.03 (0.02)	0.03 (0.02)	0.02 (0.01)	0.02 (0.02)	0.03 (0.02)
Ang ML (°)	2.53 (1.85)	2.59 (1.88)	2.33 (1.47)	1.93 (1.10)	1.75 (1.36)	2.21 (1.47)
ML range (m)	0.02 (0.01)	0.02 (0.01)	0.02 (0.01)	0.01 (0.01)	0.01 (0.01)	0.01 (0.01)

## Data Availability

Not applicable.

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
