# Peer review of "Influence of Foot Morphology on the Center of Pressure Pattern in Patients with Down Syndrome"

_ijerph, 2023, doi:10.3390/ijerph20042769_

Round 1

Reviewer 1 Report

January 14, 2023

Review of Ijerph-2160951

Gait analysis of patients with Down Syndrome: influence of foot morphology on Centre of pressure.

The manuscript reported on a study that examined the relationship between foot morphology and gait in individuals with Down syndrome (DS).  Gait analysis focused on the center (centre?) of pressure (CP).  Differences were found between participants with DS and healthy controls.  The manuscript also reported differences between children and adults with DS. The CP characteristics in participants with DS were judged to negatively affect mobility.  Speculation as to the origin of the gait characteristics recorded in those with DS were offered.

Introduction:

The introduction has two major flaws.  First, the introduction does not offer a compelling rationale for the study.  The authors note that little information is available regarding the trajectory of the CP during the gait of individuals with DS.  This is the only rationale the authors offer for conducting the study, and it is insufficient.  Not everything little is known about is worth studying.  The authors need to provide a detailed and convincing rationale for devoting effort to examining the trajectory of the CP during the gait of individuals with DS.  Why should anyone care about the existence of a possible link between flat feet and alterations in the trajectory of the CP during walking?  In short, why is it important to study this topic??

The second major flaw of the introduction is that it offers no hypotheses whatsoever.  Without a detailed accounting of what the researches expected to find in their study, the study appears to be a fishing trip, and this makes the remainder of the manuscript is difficult to evaluate.  The methods, results and discussion sections of a manuscript should all emanate from the hypotheses offered in the introduction.

Methods:

L63:  The initial section of the Methods seems misplaced.  Would make more sense to move this content closer to L121.

L 83:  The description of participants [65 children (<12 years), and 152 adolescents and young adults (>12 years)], differs from what is presented in the abstract (i.e., 65 children and 152 adults). It appears that the authors actually categorized participants 12-17 years of age as adults.  This is not appropriate.  I am not aware of any reasonable definition of adulthood that is lower than 16 years of age.  There is a consensus that adulthood can be said to begin at 18 years of age.  I think it will be essential for the authors to conduct a new analysis using the categories of children, adolescents and adults.

L 103: Could participants see the force plate?  If so, were they instructed to step on it as they walked? 

How many trials were used to generate CP data? Six?

Results:

Tables are cluttered and I found myself desirous of a graphic presentation of the most salient results.

Discussion

L273-282. Duplicate paragraphs??

Two factors prevented me from seriously evaluating the results and discussion.

First, the authors offered no hypotheses in the introduction, and second, the age categorization of their participants is dubious. 

I would recommend revising the manuscript to include a compelling rational for the study, as well as detailed hypotheses.  Also, I strongly recommended the authors conduct a new analysis of their data using three age categories, children, adolescents and adults.

Author Response

Response to Reviewer  

  

Dear reviewer,  

As a premise, we are grateful to you for the time you have devoted to our work, for your general judgement and for your useful suggestions aimed to improve the quality of the paper. We have tried to acknowledge them in the revised paper point-by-point as described in the following, hoping the final work will improve.  All the revised parts of the original manuscript have been highlighted in order to facilitate their detection.  

Thank you.  

Referee: 1

The manuscript reported on a study that examined the relationship between foot morphology and gait in individuals with Down syndrome (DS).  Gait analysis focused on the center (centre?) of pressure (CP).

All ''centre'' have been changed to ''center,'' as suggested

Differences were found between participants with DS and healthy controls.  The manuscript also reported differences between children and adults with DS. The CP characteristics in participants with DS were judged to negatively affect mobility.  Speculation as to the origin of the gait characteristics recorded in those with DS were offered.

Introduction:

The introduction has two major flaws.  First, the introduction does not offer a compelling rationale for the study.  The authors note that little information is available regarding the trajectory of the CP during the gait of individuals with DS.  This is the only rationale the authors offer for conducting the study, and it is insufficient.  Not everything little is known about is worth studying.  The authors need to provide a detailed and convincing rationale for devoting effort to examining the trajectory of the CP during the gait of individuals with DS.  Why should anyone care about the existence of a possible link between flat feet and alterations in the trajectory of the CP during walking?  In short, why is it important to study this topic??

The importance of the study has been better specified in the text, as suggested

The second major flaw of the introduction is that it offers no hypotheses whatsoever.  Without a detailed accounting of what the researches expected to find in their study, the study appears to be a fishing trip, and this makes the remainder of the manuscript is difficult to evaluate.  The methods, results and discussion sections of a manuscript should all emanate from the hypotheses offered in the introduction.

The introduction has been revised and expanded following these suggestions.

Methods:

L63:  The initial section of the Methods seems misplaced.  Would make more sense to move this content closer to L121.

As you suggested, we noticed that the initial part of the materials and methods was not consistent in this section. Since it is the explanation of AI as defined in the literature, it was decided to move it to the introduction.

 L 83:  The description of participants [65 children (<12 years), and 152 adolescents and young adults (>12 years)], differs from what is presented in the abstract (i.e., 65 children and 152 adults). It appears that the authors actually categorized participants 12-17 years of age as adults.  This is not appropriate.  I am not aware of any reasonable definition of adulthood that is lower than 16 years of age.  There is a consensus that adulthood can be said to begin at 18 years of age.  I think it will be essential for the authors to conduct a new analysis using the categories of children, adolescents and adults.

We thank you for your comment but unfortunately the division into adolescents and adults could not be made due to the small number of participants in the control group (n = 11). The adult group was renamed as the young adult group given the overall young age of the whole group.

 L 103: Could participants see the force plate?  If so, were they instructed to step on it as they walked?

Participants were not given any instructions on how to walk on the force platforms to prevent affecting the test results. The only instruction given was to walk at a comfortable self-selected speed, as specify in the text.

 How many trials were used to generate CP data? Six?

Yes, to avoid misunderstanding, the sentence in the text specifying the number of trials has been changed.

Results:

Tables are cluttered and I found myself desirous of a graphic presentation of the most salient results.

 We thank you for the comment but we believe that a graphical representation does not add value over tables, being a histogram representation of the mean value and the standard deviation.

 Discussion

L273-282. Duplicate paragraphs??

The paragraph has been removed; we apologize for the oversight.

 Two factors prevented me from seriously evaluating the results and discussion.

First, the authors offered no hypotheses in the introduction, and second, the age categorization of their participants is dubious.

I would recommend revising the manuscript to include a compelling rational for the study, as well as detailed hypotheses.  Also, I strongly recommended the authors conduct a new analysis of their data using three age categories, children, adolescents and adults.

We thank for the comments and revisions made.

Reviewer 2 Report

Thank you for opportunity for reviewing this interesting paper. The research adhere to reporting strobe guidelines. After carefully reading this manuscript, I must say that, from my point of view, the authors have done research on an important topic related with the importance of the  gait analysis of patients with down syndrome. This could be interesting clinicians, universities, private research organizations, and independent scientists, that frequently work in this area. It could give them a wider concept about and helps advance recognition of the input of different health professionals into the management of this condition, and helps inform the need for further multi-professional work in this area. This is an interesting aim  to assess how the different conformation of the foot in  individuals with Down Syndrome affects the CoP during walking . I have considered the quality of the manuscript redaction and presentation, the quality of the research methodology, the novelty and importance of the observations, and the appropriateness for the Journal’s readers (according with the Journal’s name). I have no real problems with the text of this paper, only some suggestions that are mentioned below. It appears as if the authors have done the study well, have answered an interesting clinical question with their work but there are a major concerns with the manuscript that require attention prior to publication. These will be discussed below relative to the sections of the manuscript. TITLE The title of this manuscript is very long. Perhaps a more concise version for clarity, interes and ease of read. KEYWORDS: Please use recognised MeSH terms as this will assist others when they are searching for information on your research topic. The following website will provide these (simply start typing in a keyword and see if it exists or find an alternative if it does not): https://www.ncbi.nlm.nih.gov/mesh ABSTRACt: It is hard to get the detail in an abstract when the word count is limited and this is often the hardest part of a paper to write. However, I do feel that it would be beneficial to explain the aim and conclusions what specifically you are looking at in relation to outcomes in this research. INTRODUCTION I suggest that background should be improved, with more details about foot Health of Persons with Down Syndrome, more info in the research of Calvo Lobo et al https://pubmed.ncbi.nlm.nih.gov/29757962/ and Cala-Perez et al https://pubmed.ncbi.nlm.nih.gov/30892480/

Thus, it is indeed important paper but it lacks several critical references, in which it was presented related with this condition, and it should be emphasized in the INTRODUCTION or Discussion of the authors' paper. METHODS This section is poor, needs to present a better rationale for the study and the methodology employed. Also, neither appear information related with inclusion and exclusion criteria, dates, protocol. RESULTS The results is clear and concise with appropriate statistical analysis been performed appropriately and rigorously. DISCUSSION. This section is very short a rehashing of the results. It does not appear that the authors include much interpretation of what the study findings mean for clinical practice or research related with other research. CONCLUSION: These conclusions need to be softened, modified a in order to reflect only the study findings.

Author Response

Response to Reviewer  

  

Dear reviewer,  

As a premise, we are grateful to you for the time you have devoted to our work, for your general judgement and for your useful suggestions aimed to improve the quality of the paper. We have tried to acknowledge them in the revised paper point-by-point as described in the following, hoping the final work will improve.  All the revised parts of the original manuscript have been highlighted in order to facilitate their detection.  

Thank you.  

Referee: 2

Thank you for opportunity for reviewing this interesting paper. The research adhere to reporting strobe guidelines. After carefully reading this manuscript, I must say that, from my point of view, the authors have done research on an important topic related with the importance of the  gait analysis of patients with down syndrome. This could be interesting clinicians, universities, private research organizations, and independent scientists, that frequently work in this area. It could give them a wider concept about and helps advance recognition of the input of different health professionals into the management of this condition, and helps inform the need for further multi-professional work in this area. This is an interesting aim  to assess how the different conformation of the foot in  individuals with Down Syndrome affects the CoP during walking . I have considered the quality of the manuscript redaction and presentation, the quality of the research methodology, the novelty and importance of the observations, and the appropriateness for the Journal’s readers (according with the Journal’s name). I have no real problems with the text of this paper, only some suggestions that are mentioned below. It appears as if the authors have done the study well, have answered an interesting clinical question with their work but there are a major concerns with the manuscript that require attention prior to publication. These will be discussed below relative to the sections of the manuscript.

We thank for the comments.

TITLE The title of this manuscript is very long. Perhaps a more concise version for clarity, interes and ease of read.

Title has been changed, as suggested.

KEYWORDS: Please use recognised MeSH terms as this will assist others when they are searching for information on your research topic. The following website will provide these (simply start typing in a keyword and see if it exists or find an alternative if it does not): https://www.ncbi.nlm.nih.gov/mesh

Keywords have been changed, as suggested.

ABSTRACt: It is hard to get the detail in an abstract when the word count is limited and this is often the hardest part of a paper to write. However, I do feel that it would be beneficial to explain the aim and conclusions what specifically you are looking at in relation to outcomes in this research.

The abstract has been revised following the suggestions.

INTRODUCTION I suggest that background should be improved, with more details about foot Health of Persons with Down Syndrome, more info in the research of Calvo Lobo et al https://pubmed.ncbi.nlm.nih.gov/29757962/ and Cala-Perez et al https://pubmed.ncbi.nlm.nih.gov/30892480/ Thus, it is indeed important paper but it lacks several critical references, in which it was presented related with this condition, and it should be emphasized in the INTRODUCTION or Discussion of the authors' paper.

The requested insights with the relevant references have been added, as suggested.

METHODS This section is poor, needs to present a better rationale for the study and the methodology employed. Also, neither appear information related with inclusion and exclusion criteria, dates, protocol.

All requested details have been added as suggested.

RESULTS The results is clear and concise with appropriate statistical analysis been performed appropriately and rigorously.

We thank for the comments.

DISCUSSION. This section is very short a rehashing of the results. It does not appear that the authors include much interpretation of what the study findings mean for clinical practice or research related with other research. CONCLUSION: These conclusions need to be softened, modified a in order to reflect only the study findings.

The discussion and conclusion have been rewritten according to the suggestions given.

Round 2

Reviewer 2 Report

The authors have clearly and adequately addressed all comments raised by the reviewers. Please also consider update the of references using the format of this journal https://www.mdpi.com/journal/ijerph/instructions

Author Response

Thank you for the comment, we confirm that we use the template given in https://www.mdpi.com/journal/ijerph/instructions.